# Non-viral *in vivo* electroporation-based chromosomal engineering and repair assessment in the murine uterine epithelium

Satoru Iwata[1,2,3,4*], Yumi Miura[1], Takashi Iwamoto[1,2]

1 Center for Education in Laboratory Animal Research, Chubu University, Kasugai, Aichi, Japan,
2 Department of Biomedical Sciences, College of Life and Health Sciences, Chubu University, Kasugai, Aichi, Japan, 3 College of Bioscience and Biotechnology, Chubu University, Kasugai, Aichi, Japan,
4 Center for Mathematical Science and Artificial Intelligence, Chubu University, Kasugai, Aichi, Japan

* satoru_iwata@fsc.chubu.ac.jp

## Abstract

Chromosomal rearrangements generated by CRISPR/Cas systems are valuable for studying genomic architecture and repair mechanisms. However, most *in vivo* approaches rely on viral vectors, which require specialised production, prolonged nuclease expression, and elevated biosafety containment. Here, we applied Cas9 ribonucleoprotein (RNP) electroporation to the murine uterine epithelium as a simple, non-viral strategy for somatic chromosomal engineering. This method successfully induced defined interchromosomal translocations at multiple loci and enabled the molecular assessment of large-scale inversion repair (57.8 Mb) using paired gRNAs with an ssODN donor. While rearranged alleles were detected at low apparent frequencies in bulk uterine DNA—consistent with epithelial-restricted delivery and somatic mosaicism—high-depth whole-genome sequencing (WGS) and PCR provided nucleotide-resolution confirmation of precise junction formation. Our findings demonstrate that uterine electroporation of CRISPR RNPs is a feasible, rapid approach for evaluating engineered chromosomal rearrangements *in vivo*, providing a controlled platform for analyzing somatic DNA repair outcomes without viral confounds.

## Introduction

Chromosomal abnormalities, particularly inversions and translocations, predominantly arise in somatic tissues where they play a central role in cancer initiation and progression. Such rearrangements can alter gene expression and promote tumourigenesis by dysregulating oncogenes and tumour suppressor genes [1,2]. Although less frequent, germline rearrangements can also occur, leading to infertility or hereditary disorders by disrupting genes essential for normal physiological functions [3–6]. Understanding the mechanisms underlying the formation and repair of these

**Data availability statement:** All relevant data are within the paper and its Supporting Information file.

**Funding:** This study was supported by the JSPS KAKENHI Grant Numbers 23K21287 and 25K23678 (to SI), the Science Research Promotion Fund from the Promotion and Mutual Aid Corporation for Private Schools of Japan (to SI), and a Chubu University Grant (S) (to SI).

**Competing interests:** The authors have declared that no competing interests exist.

abnormalities is crucial, and the clustered regularly interspaced palindromic repeat (CRISPR)/CRISPR-associated protein (Cas) system has become a widely used tool for modelling and investigating such events [7,8].

The CRISPR/Cas system has been successfully employed to generate model organisms with specific chromosomal abnormalities [9–11], contributing significantly to the elucidation of disease mechanisms. Recently, advanced prime editing techniques, such as prime editing with wild-type Cas9 (WT-PE) and prime editor nuclease-mediated translocation and inversion (PETI), have exhibited high precision in inducing rearrangements in cultured human cells [12,13]. However, despite these *in vitro* advances, the induction of defined chromosomal rearrangements directly in somatic tissues *in vivo* remains technically challenging.

Current *in vivo* approaches to induce chromosomal rearrangements often rely on viral vectors (e.g., AAV, Lentivirus). While effective for delivery, viral methods present challenges such as cargo size limitations, immunogenicity, and prolonged nuclease expression. Constitutive expression of nucleases can lead to recurrent cleavage and off-target effects, which may obscure the analysis of primary DNA repair outcomes. Consequently, there is a need for non-viral alternatives that allow for transient expression and "clean" delivery profiles suitable for somatic tissues. Our objective is to establish a simple and controlled non-viral platform to induce and evaluate defined chromosomal rearrangements, serving as a complementary approach to robust viral systems.

We have previously demonstrated the potential of *in vivo* electroporation by delivering CRISPR RNPs to fertilised mouse embryos to induce complex chromosomal rearrangements [14]. However, the widespread application of embryo-based approaches is constrained by ethical regulations regarding human embryo editing and technical restrictions on the use of fertilised eggs [15]. Beyond these limitations, a critical question remains regarding the fidelity of DNA repair in somatic tissues. In our previous embryo studies involving *Recql5* mutations, we observed that repair frequently involved replication-based complex mechanisms, such as microhomology-mediated break-induced replication (MMBIR) and template switching (FoSTeS), leading to unintended structural alterations [16]. It remains unclear whether targeted chromosomal breaks in wild-type somatic tissues are repaired through similar error-prone mechanisms or via precise ligation. This necessitates the development of a somatic validation system to rigorously evaluate repair outcomes.

In this study, we extended the scope of chromosomal engineering to somatic organ tissues using *in vivo* electroporation as a proof-of-concept. Adapting the CRISPR RNP electroporation method developed by Kobayashi et al. [17], we examined its capability to induce defined chromosomal rearrangements. Although the oviductal epithelium has been widely used for genome editing via intraoviductal electroporation [18,19], here we focused on the murine uterine epithelium—a tissue of origin for uterine cancers often harbouring inversions and translocations [20]. Crucially, we utilized this system to assess the fidelity of somatic DNA repair at nucleotide resolution, specifically investigating whether FoSTeS/MMBIR signatures occur during the repair of large-scale inversions in a somatic context. Our findings

demonstrate that *in vivo* electroporation is a feasible, virus-free strategy for generating and evaluating engineered chromosomal rearrangements in the uterine epithelium.

## Materials and methods

### Experimental animals

Wild-type (WT) mice (C57BL/6NCrSlc; Japan SLC, Shizuoka, Japan) and *In(6)1J* inversion-carrying mice (C3H/HeJJcl; CLEA Japan, Tokyo, Japan) were used. Sexually mature female mice aged 8 weeks or older were selected for experiments. All animals were maintained under controlled environmental conditions at a constant temperature of 22±2 °C and humidity of 50±10% under a 12/12-h light/dark cycle. All animal experiments were approved by the Institutional Animal Care and Use Committee of Chubu University (permit number #202410012; Kasugai, Aichi, Japan) and adhered to the relevant guidelines.

### CRISPR RNP and single-stranded oligodeoxynucleotide (ssODN) preparation

CRISPR guide RNAs (gRNAs) were designed using CHOPCHOP (http://chopchop.cbu.uib.no/; S1 Table) [21]. The CRISPR RNP complex was assembled using the Alt-R S.p. Cas9 Nuclease 3NLS (Integrated DNA Technologies, Coralville, IA, USA) and a custom-designed gRNA consisting of a CRISPR RNA (crRNA): transactivating CRISPR RNA (tracrRNA) duplex (Integrated DNA Technologies). After resuspending in the nuclease-free duplex buffer (Integrated DNA Technologies) at a final concentration of 4,000 ng/µL, crRNA and tracrRNA were mixed at equimolar ratios, heated at 95 °C for 10 min, and gradually cooled to 25 °C to facilitate duplex formation. The crRNA:tracrRNA duplex was then incubated with Alt-R S.p. Cas9 Nuclease 3NLS at 25 °C for 10 min to assemble the RNP complex. Specific concentrations are indicated in the following section: *in vivo* genome editing of murine endometrium. Subsequently, ssODNs were synthesised by Eurofins Genomics (Tokyo, Japan) to facilitate the seamless joining of the two DNA sequences, ensuring that the junction was positioned at the centre of the predicted cleavage sites located within 3 bp of the PAM sequences (S2 Table). Moreover, 5′ and 3′ ends of ssODNs were modified with two consecutive phosphorothioate (*) linkages to enhance the HDR efficiency [22] (S2 Table).

### *In vivo* genome editing of the murine endometrium

Female metestrus or diestrus mice were subjected to *in vivo* genome editing of the murine endometrium using a modified version of the method described by Kobayashi et al. [17]. This procedure targets the uterine epithelium and is conducted postnatally; therefore, genome editing reagents do not access the ovaries, and oocytes remain unedited. These stages were selected because the uterine horns are long, flaccid, and relatively thick-walled during metestrus and diestrus, which facilitates intrauterine procedures. In contrast, during oestrus, the uterus is thin-walled, nearly transparent, fluid-filled, and turgid, making surgical manipulation more difficult and less reproducible [23]. To generate complex CCRs, the following CRISPR solution was used: 520 ng/µL Alt-R S.p. Cas9 Nuclease 3NLS, 30 µM crRNA:tracrRNA duplex targeting the three designed junction sites (*Hmga2–Wif1*, *Hmga2–Rassf3*, and *Wif1–Rassf3*), with 158 ng/µL ssODN for each corresponding junction site, and 0.02% Fast Green FCF (Wako, Osaka, Japan) marker diluted in the Opti-MEM (Thermo Fisher Scientific, Waltham, MA, USA). To generate chromosomal inversions and translocations, the following CRISPR solution was used: 520 ng/µL Alt-R S.p. Cas9 Nuclease 3NLS, 30 µM crRNA:tracrRNA duplex targeting the two flanking sites for inversions (left and right cut sites on the same chromosome) and each specific breakpoint on the respective chromosomes for translocations, 158 ng/µL ssODN corresponding to each targeted site, and 0.02% Fast Green FCF (Wako) marker diluted in the Opti-MEM (Thermo Fisher Scientific). Comparative experiments were conducted by delivering the CRISPR/Cas9 RNPs with or without ssODNs under identical electroporation conditions. Each condition was tested in 2–3 independent trials using different animals to ensure reproducibility. Before electroporation, the female mice were anaesthetised with a

mixture of medetomidine (0.75 mg/kg, Nippon Zenyaku Kogyo, Fukushima, Japan), midazolam (4 mg/kg; Sandoz, Tokyo, Japan), and butorphanol (5 mg/kg; Meiji Seika Pharma, Tokyo, Japan).

Glass micropipettes were pulled using a vertical capillary puller (NARISHIGE, Tokyo, Japan) to create fine-tipped capillaries suitable for microinjection. Before use, the tip of each capillary was cut diagonally to a length of 0.5 mm using microscissors to facilitate smooth reagent flow. The cut capillary was then connected to a syringe via a silicone tube and filled with CRISPR solution to ensure complete reagent loading. Once filled, the capillary was attached to a holder and secured in place using a flexible stand. The CRISPR solution (2 μL) was then introduced into the uterine lumen via microinjection through the uterotubal junction using the prepared glass micropipette and the FemtoJet 4i microinjector (Eppendorf, Hamburg, Germany). Injection was performed under the following parameters: injection pressure (pi) = 100 hPa, injection duration (ti) = Manual, and compensation pressure (pc) = 10 hPa. To prevent reagent backflow, the oviduct near the ovary was clamped with a haemostatic clip (Natsume Seisakusho, Tokyo, Japan). The uterus was clamped to the cervical side by using a haemostatic clip (Natsume Seisakusho) to prevent leakage. Following injection, the uterine horns were clamped using tweezer electrodes (LF650P5; BEX, Tokyo, Japan), and electroporation was performed using a CUY21EDIT II electroporator (BEX). The electrode orientation was adjusted, and three additional sets of electric pulses were applied. The following electroporation parameters were used: primary large pulse (PLP): 20 V, 30.0-ms pulse duration, 50.0-ms pulse interval, 3 pulses, 10% decay (± pulse orientation). Secondary decay pulse (Pd): 10 V, 50.0-ms pulse duration, 50.0-ms pulse interval, 3 pulses, 40% decay (± pulse orientation). After electroporation, the uterine horns were repositioned, and the incisions were sutured. To reverse the effects of medetomidine, atipamezole hydrochloride (0.75 mg/kg, Nippon Zenyaku Kogyo, Fukushima, Japan) was intraperitoneally administered postoperatively. Following surgery, mice were placed on a warming pad maintained at 38°C overnight to support recovery and minimize postoperative suffering, in accordance with the 3Rs principles. In total, 50 mice were used across all genome editing experiments. At the experimental endpoint (≥7 days post-electroporation), mice were euthanised by cervical dislocation.

### Fluorescent tracer delivery and tissue processing for spatial localization

To assess the spatial distribution of electroporated material, Dextran-Tetramethylrhodamine (TMR; 3,000 MW, anionic, lysine-fixable; 2 μg/μL) was introduced into the uterine lumen using the same microinjection and electroporation procedures described above. Three mice were euthanised by cervical dislocation at 1-hour post-electroporation, and the uterine horns were dissected under reduced light. Tissues were fixed in 4% paraformaldehyde in PBS (pH 7.4) at room temperature for 48 h, washed once in PBS for 60 min, and then washed twice in 70% ethanol (60 min each). Fixed tissues were dehydrated, paraffin-embedded, and sectioned at 5–7 μm; the slides were prepared by Morphotechnology Co., Ltd. (Sapporo, Japan). Sections were stained with DAPI (Invitrogen, Carlsbad, CA, USA) at room temperature for 1 min, briefly rinsed in PBS, and mounted. Fluorescent images were acquired using a NanoZoomer virtual slide scanner (Hamamatsu Photonics, Hamamatsu, Japan) with dextran TMR and DAPI filter sets.

### Analysis of the CRISPR/Cas9-engineered mice

To screen for CRISPR/Cas9-induced mutations, genomic DNA was extracted from the uterine horns of edited mice using a DNeasy Blood & Tissue Kit (Qiagen, Hilden, Germany) and subjected to polymerase chain reaction (PCR) amplification using EmeraldAmp PCR Master Mix (Takara Bio, Shiga, Japan). During dissection, the entire uterine horn was excised without separating the tissue layers. Genomic DNA was extracted from whole-thickness uterine tissue, including the epithelial layer (which is the primary target of electroporation) and the underlying stromal and muscular layers, which were not expected to be transfected. This comprehensive whole-organ approach was selected to mitigate potential sampling bias that may arise due to the region-dependent variability of *in vivo* electroporation. By avoiding the collection of solely highly or poorly edited regions, we aim to ensure that the observed outcomes accurately reflect the spatially averaged effects, accepting the trade-off of signal dilution by non-target stromal tissues.

For nested PCR, the first round consisted of 40 cycles of denaturation at 98 °C for 10 s, annealing at 60 °C for 30 s, and extension at 72 °C for 1 min, followed by a final indefinite hold at 10 °C. The second round of PCR consisted of 15 cycles of denaturation at 98 °C for 10 s, annealing at 63 °C for 30 s, and extension at 72 °C for 30 s, with a final indefinite hold at 10 °C. The resulting PCR products were purified using the NucleoSpin Gel and PCR Cleanup Kit (Takara Bio) and cloned into the pTAC-1 vector (Biodynamics, Tokyo, Japan). Individual clone sequences were determined using Sanger sequencing (Eurofins Genomics). To minimise the possibility of recombination events occurring during bacterial propagation, PCR products were cloned into a TA vector and transformed into chemically competent *E. coli* (DH5α), with minimal outgrowth time before plasmid isolation. Recombination junctions were consistently observed across multiple independent clones from different animals, reducing the likelihood that they arose as bacterial artefacts. All PCR primers used for genotyping are listed in S3 Table. To evaluate the chromosomal rearrangements, four colonies per uterine sample were analysed by colony PCR using breakpoint-specific primer sets. A sample was considered positive for a given junction if at least one of four clones contained the corresponding ligation.

The repair outcomes were classified based on sequence features at the breakpoint junctions. When a single-stranded oligodeoxynucleotide (ssODN) donor was used and the repair junction precisely matched the donor template sequence, the event was classified as homology-directed repair (HDR). However, in some cases, it may also reflect the accurate re-ligation of Cas9-generated ends without end processing. Junctions lacking sequence homology were classified as NHEJ, whereas those containing microhomology sequences of two base pairs (bp) or more were classified as MMEJ, consistent with commonly accepted definitions [24].

## Whole-genome sequencing and structural-variant analysis

Total genomic DNA was extracted from the electroporated uterine horns using a DNeasy Blood & Tissue Kit (Qiagen, Hilden, Germany) per the manufacturer's protocol. Libraries were prepared using the TruSeq Nano DNA Library Prep Kit (Illumina, San Diego, CA, USA), and sequencing was performed by Macrogen Japan Inc. (Tokyo, Japan) on an Illumina NovaSeq6000 platform, generating 150 bp paired-end reads (approximately 90 Gb per sample).

Sequencing reads were aligned to the GRCm38/mm10 mouse reference genome using standard pipelines to generate the BAM files. For structural-variant interrogation at the intended loci, BAM files were queried using samtools v1.22.1 [25]. Candidate regions were examined within ±2 Mb windows centered on the predicted breakpoints. Low-quality, duplicate, and supplementary alignments were excluded using the parameters -q 20 -F 0x904.

To extract discordant read pairs linking candidate loci, the SAMtools view was combined with AWK filters on the mate chromosome and position. Read names of discordant pairs were then retrieved, and full alignments were exported using Samtools view (with -h and -N) for visualization. Visualization was performed using Integrative Genomics Viewer (IGV, v2.19.5) [26]. The reference coordinates were consistently based on mm10 throughout all analyses.

To obtain a minimal quantitative estimate of junction-supporting reads in bulk uterine tissue, we calculated the apparent editing frequency at each locus by dividing the number of discordant paired-end reads (PE-support) with the number of properly paired reads within the corresponding ±2 Mb window (PPmean). This simple ratio (PE-support/PPmean) reflects the relative abundance of rearranged alleles in the sequenced tissue, without assuming a clonal structure or purity.

The WGS data reported in this paper were deposited into the DNA Data Bank of Japan Sequence Read Archive (https://ddbj.nig.ac.jp/search/entry/bioproject/PRJDB37706).

## Results

### Feasibility of *in vivo* editing for CCR in the mouse uterine epithelium

To investigate the feasibility of inducing defined chromosomal rearrangements in somatic tissues, we adapted the CRISPR RNP electroporation method developed by Kobayashi et al. [17] for the murine uterine epithelium (Fig 1A).

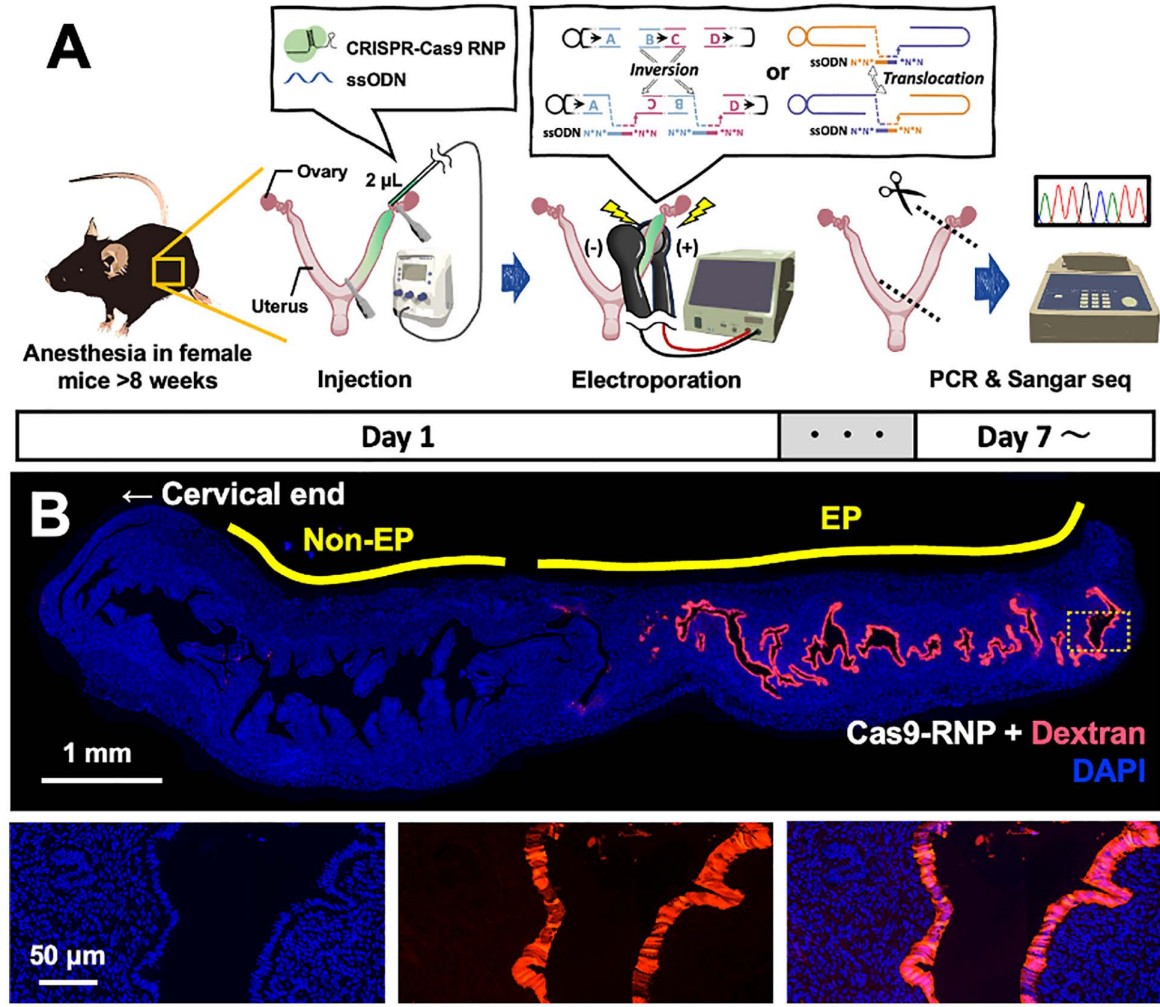

**Fig 1.** *In vivo* **electroporation-based chromosomal engineering in the murine uterus. (A)** The experimental workflow. Anaesthetised female mice (>8 weeks old) received intrauterine injection of CRISPR–Cas9 RNP with or without ssODN donor (2 µL), followed by electroporation across the uterine horns. Genomic DNA collected ≥7 days post-electroporation for PCR and Sanger sequencing. **(B)** Distribution of the electroporated material. Dextran (red) was co-delivered with Cas9–RNP. Non-EP, non-electroporated region (cervical side); EP, electroporated region (ovarian side). Nuclei were counter-stained with DAPI (blue). The arrow indicates the cervical end.

CRISPR RNPs were co-injected with single-stranded oligodeoxynucleotides (ssODNs) designed to bridge the predicted breakpoints and guide the repair of prespecified junctions. The mixture was introduced into the uterine lumen via a micro-injector to minimize mechanical stress and leakage, followed by electroporation into the epithelial cells. The inclusion of ssODNs served to promote precise junction formation via homology-directed repair (HDR), allowing us to assess whether donor templates could facilitate accurate ligation at chromosomal breakpoints.

To verify the spatial distribution of reagent delivery prior to evaluating genome editing outcomes, we co-introduced a fluorescent dextran tracer during *in vivo* electroporation. Paraffin sections of uterine tissues revealed strong fluorescence signals confined to the uterine epithelium in the electroporated region (EP), whereas no detectable signal was observed in the non-electroporated region (Non-EP) on the cervical side, despite both regions being filled with the injected solution (Fig 1B). This confirms that dextran uptake is electroporation-dependent and supports the expectation of mosaic editing within the tissue, consistent with previous reports using this approach [17].

As an initial validation target, we selected a 1.1-Mb region on mouse chromosome 10, encompassing the *high mobility group AT-hook 2* (*Hmga2*), *Wnt inhibitory factor 1* (*Wif1*), and *Ras association domain family member 3* (*Rassf3*) genes. This locus is implicated in various human cancers and shares a syntenic configuration with human chromosome 12 [27] (Fig 2A), and was previously targeted in our embryo-based studies [14]. The simultaneous use of three cut sites was intentionally designed to recapitulate multi-breakpoint CCRs and to assess whether our electroporation platform can handle such complexity in a somatic context. At postoperative day 7, genomic DNA was extracted from the whole uterine tissue, and the induction of chromosomal rearrangements was assessed via polymerase chain reaction (PCR) and Sanger sequencing.

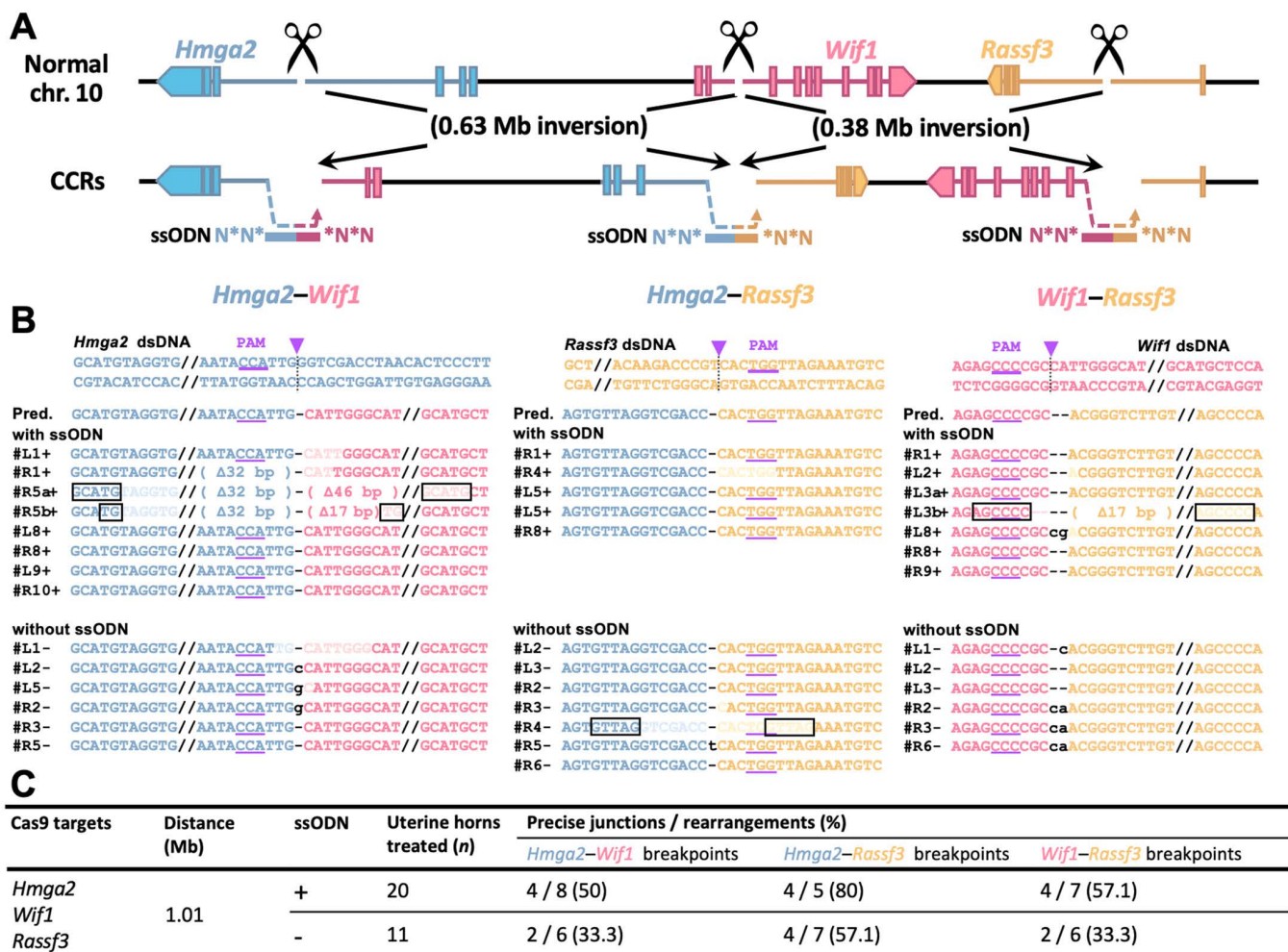

**Fig 2. CCRs in the uterine epithelium induced via *in vivo* electroporation. (A)** Schematic of CCR design involving *Hmga2*, *Wif1*, and *Rassf3* on chromosome 10, showing Cas9 cut sites (scissors) and ssODN donors for precise junction repair. **(B)** Sequence alignment of PCR products corresponding to the genomic breakpoint junctions, *Hmga2–Wif1*, *Hmga2–Rassf3*, and *Wif1–Rassf3*. Samples are grouped as "with ssODN" and "without ssODN"; "+" and "−" after sample numbers indicate the respective condition. Arrows indicate Cas9 cleavage sites; boxed bases show microhomology; lowercase letters, insertions or mismatches; underlines, PAM sequences. PAM, protospacer adjacent motif; dsDNA, double-stranded DNA; Pred., predicted sequence. **(C)** CCR induction efficiency in the uterine epithelium with or without ssODN. Precise junctions denote ssODN-consistent joins; rearrangements, all detected junctions.

Chromosomal rearrangements were detected at comparable frequencies across the three breakpoints under both ssODN(+) and ssODN(−) conditions. In the ssODN(+) group, precise junctions were observed in 4/8 (50%) of rearrangement-positive samples at *Hmga2–Wif1*, 4/5 (80%) at *Hmga2–Rassf3*, and 4/7 (57.1%) at *Wif1–Rassf3* (Fig 2B, 2C; S1A Fig; S1 File). In comparison, the ssODN(−) group exhibited slightly lower proportions of precise junctions: 2/4 (50%), 4/7 (57%), and 2/6 (33%), respectively (Fig 2B, 2C; S1A Fig). These results indicate that while the addition of ssODNs did not increase the overall detection rate of rearrangements, it contributed to a higher proportion of precise junctions at specific loci. Importantly, non-precise junctions were characterized exclusively by small deletions (ranging from one to several tens of base pairs) or simple insertions, with no complex indels observed (Fig 2B). Given their utility in guiding precise ligation, ssODNs were consistently included in all subsequent experiments.

### Induction of interchromosomal translocations in the mouse uterine epithelium

Next, we evaluated the feasibility of inducing translocations, in which segments from two distinct chromosomes are exchanged. To select robust target loci, we focused on the *functional intergenic repeating RNA element* (*Firre*) gene located on the X chromosome. The *Firre* locus plays a central role in nuclear organization by facilitating gene activation through three-dimensional spatial interactions with regulatory regions on other chromosomes [28] (Fig 3A). In this study, we selected *Firre*-interacting regions specifically as molecular benchmarks; their known spatial proximity makes them ideal candidates to test the capability of our approach to induce interchromosomal events, rather than for investigating their biological function per se. Consequently, we did not evaluate the functional consequences of these modifications, reserving such analyses for future studies.

We designed experiments to induce translocations between the *eukaryotic translation elongation factor 1 alpha 1* (*Eef1a1*) neighbourhood locus (*Eef1a1N*) on chromosome 9 and the *activating transcription factor 4* (*Atf4*) neighbourhood locus (*Atf4N*) on chromosome 15, both of which are known to contact the *Firre* region. Additionally, we targeted transloca-tions between the *yippee like 4* (*Ypel4*) neighbourhood locus (*Ypel4N*) on chromosome 2 and *Atf4N* on chromosome 15, based on their previously described nuclear proximity to *Firre* (Fig 3A, 3C). On postoperative day 7, the predicted transloca-tion junctions were successfully detected in uterine samples by PCR (Fig 3B, 3D, 3E; S1B Fig; S2 File). These results demonstrate that interchromosomal rearrangements can be generated *in vivo* using this electroporation-based approach. Given the limited number of recoverable rearrangement-positive clones per locus, the proportions of precise junctions shown in Fig 3E are presented as qualitative indicators of repair outcome categories, consistent with the proof-of-concept scope of this study.

### Evaluation of repair fidelity during the reversion of the 57.8-Mb inversion, *In(6)1J*, in the mouse uterine epithelium

Having demonstrated the induction of targeted rearrangements, we next utilized a large-scale structural variant to investi-gate the limits and fidelity of somatic chromosomal repair. We selected the *In(6)1J* inversion on chromosome 6 as a chal-lenging model substrate. Spanning approximately 57.8 Mb (roughly 40% of chromosome 6) [29–31], this inversion offers a unique opportunity to assess whether somatic cells can precisely re-ligate chromosomal ends separated by megabase-scale distances. While *In(6)1J* is associated with skeletal phenotypes [30], we focused here on its genomic architecture to evaluate the molecular characteristics of repair junctions formed during targeted reversion.

Guide RNAs were designed near both breakpoints, and ssODNs corresponding to the wild-type genomic sequences were co-delivered to guide and analyze the precision of the repair (Fig 4A). At the left breakpoint, the repair ssODN tem-plate was based on the wild-type sequence but also contained the protospacer adjacent motif (PAM) and upstream nucle-otides matching the Cas9 target site in the *In(6)1J* genome. Due to mismatches introduced in the protospacer sequence upon repair, Cas9 activity at the restored site is expected to be substantially reduced, thereby minimizing the risk of re-cleavage [32,33]. Although RNPs degrade relatively quickly, transient delivery further mitigates the risk of re-cutting by limiting the duration during which additional cleavage can occur.

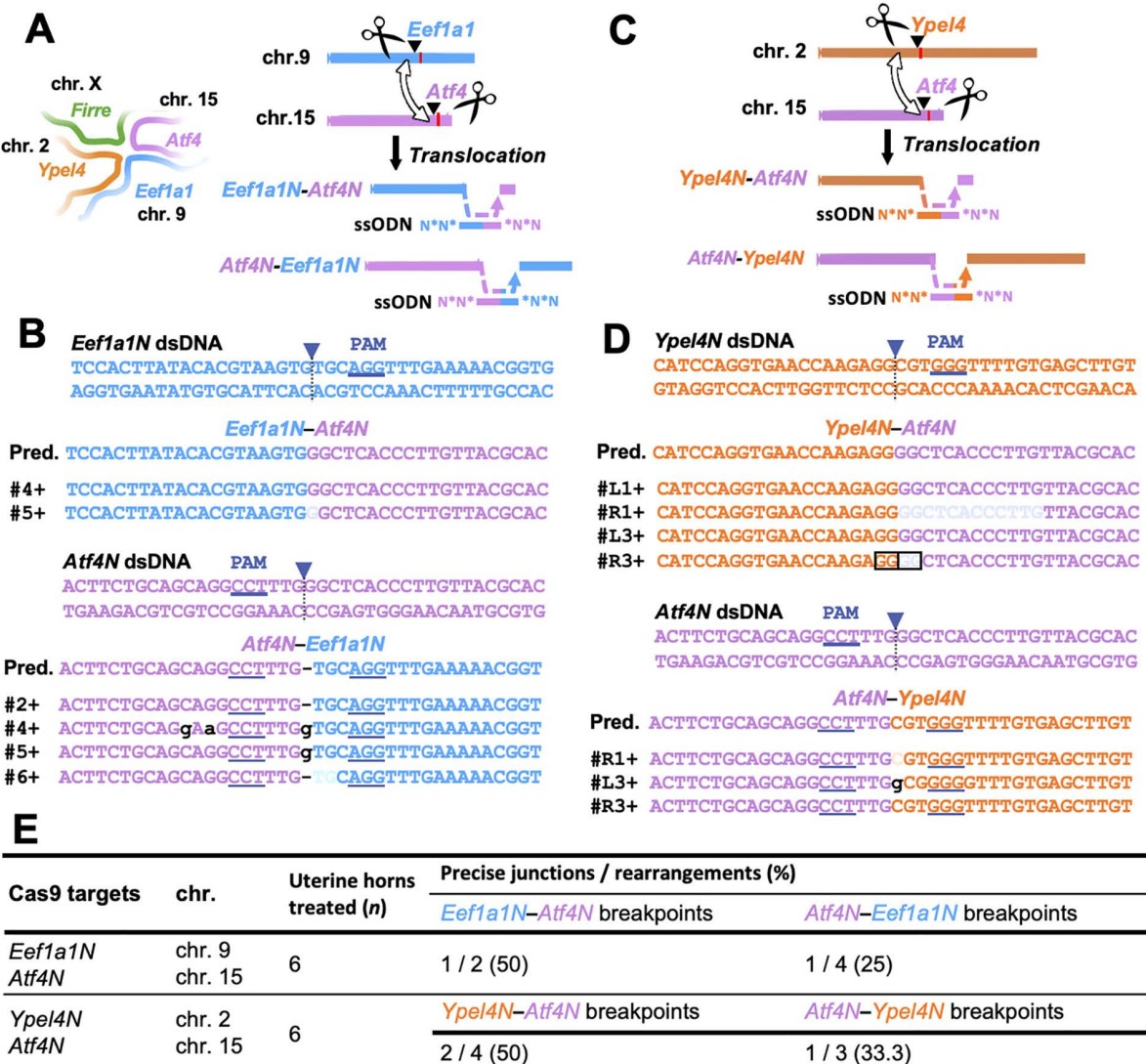

**Fig 3. Translocations at the _Firre_ locus in the uterine epithelium induced via _in vivo_ electroporation. (A)** Schematic showing the _Firre_ locus on the X chromosome. Translocation design between _Eef1a1N_ (chr. 9) and _Atf4N_ (chr. 15) is shown, with Cas9 cut sites (scissors) and ssODN donors for junction repair. **(B)** Sequence alignment of the PCR products corresponding to the genomic breakpoint junctions of _Eef1a1N_ and _Atf4N_. The arrows indicate Cas9 cleavage sites; lowercase letters, insertions or mismatches; underlines, PAM sequences. PAM, protospacer adjacent motif; dsDNA, double-stranded DNA; Pred., predicted sequence. **(C)** Schematic representation of the translocation between the _Ypel4N_ (chr. 2) and _Atf4N_ (chr. 15). **(D)** Sequence alignment of the PCR products corresponding to the genomic breakpoint junctions of _Ypel4N_ and _Atf4N_. Boxed regions indicate the microhomology sequences. **(E)** Translocation efficiency of _Eef1a1N–Atf4N_ and _Ypel4N–Atf4N_ in the uterine epithelium. The precise junctions denote ssODN-consistent joins; rearrangements, all detected junctions connecting the intended loci.

Sequence analysis revealed that the native genomic configuration could be recovered, allowing us to characterize the junction fidelity (Fig 4B, 4C). Under ssODN(+) conditions, precise restoration was confirmed in 1 of 2 sequenced amplicons at the left breakpoint and 3 of 4 at the right breakpoint. In contrast, while bridging of the breakpoints was detected in ssODN(−) experiments, they lacked the precise sequence restoration observed with donor templates (Fig 4B, 4C; S1C Fig; S3 File). Although the number of sequence-verified clones is limited, the complete absence of precise junctions in

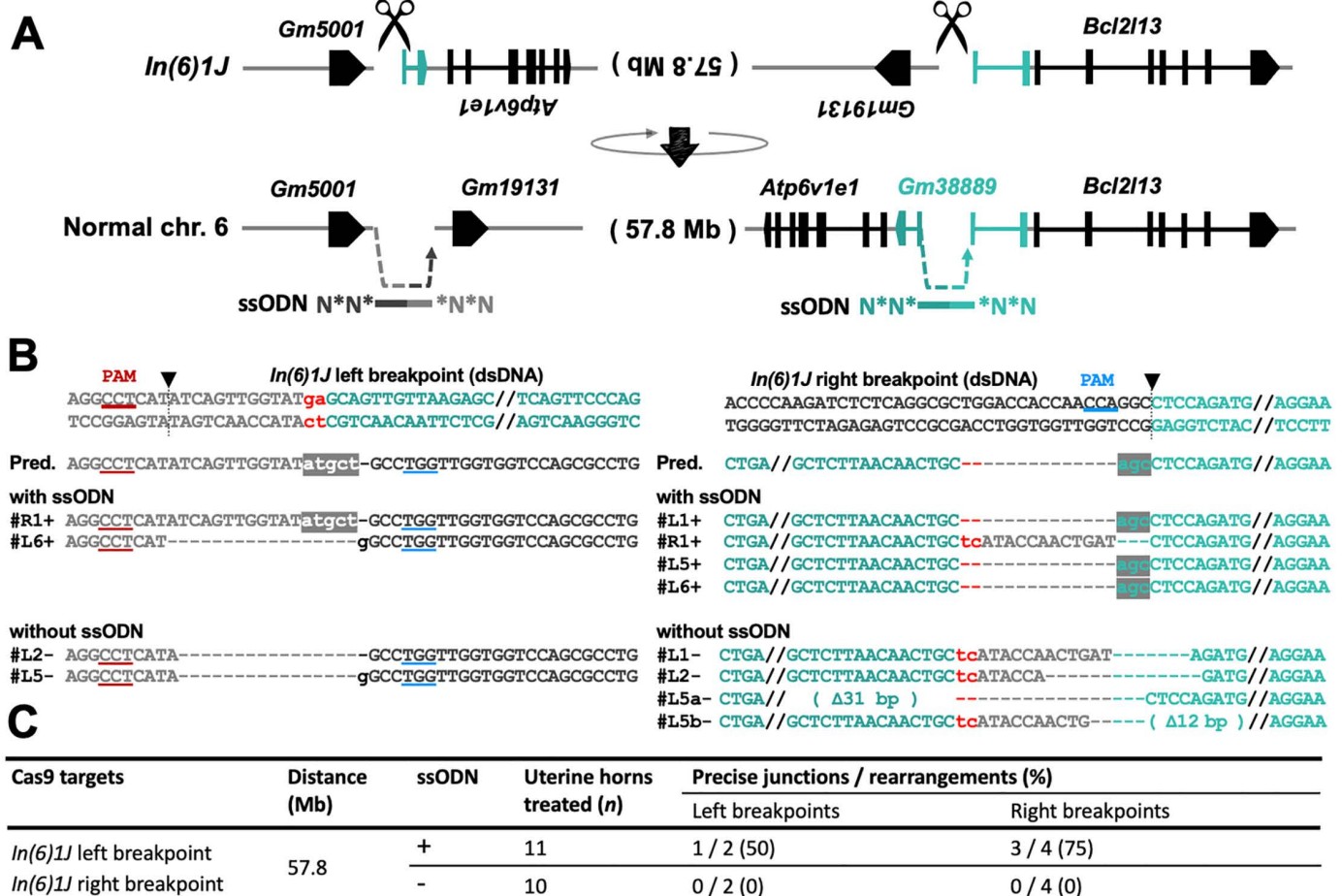

**Fig 4. Repair of a large-scale chromosomal inversion in the uterine epithelium via *in vivo* electroporation. (A)** Schematic of the CRISPR/Cas9-based strategy to repair the 57.8-Mb inversion *In(6)1J* on chromosome 6, showing Cas9 cut sites (scissors) and ssODN donors for junction repair. **(B)** Sequence alignment of the PCR products corresponding to the left and right breakpoint junctions after repair. Samples are grouped as "with ssODN" and "without ssODN"; "+" and "–" after sample numbers indicate the respective condition. Arrows mark Cas9 cleavage sites; boxed or shaded bases, microhomology or repaired regions; lowercase letters, insertions or mismatches; underlines, PAM sequences. PAM, protospacer adjacent motif; dsDNA, double-stranded DNA; Pred., predicted sequence. **(C)** Repair efficiency of *In(6)1J* in the uterine epithelium with or without ssODN. Precise junctions denote ssODN-consistent joins; rearrangements, all detectable junctions connecting the intended loci.

ssODN(−) experiments suggests that the provision of ssODNs can steer the repair machinery towards precise ligation at these distant breakpoints.

The successful detection and characterization of these 57.8-Mb reversion events demonstrate that our *in vivo* electroporation platform is capable of capturing and resolving the molecular outcomes of ultra-long-range chromosomal interactions. To the best of our knowledge, this represents the characterization of one of the largest chromosomal rearrangements targeted in a mammalian somatic tissue, providing a benchmark for assessing repair fidelity in extensive structural variants.

## Assessment of repair signatures and generalizability across multiple disease-associated loci

To further evaluate the applicability and repair fidelity of our *in vivo* electroporation approach, we targeted additional genetic loci associated with disease. Specifically, we attempted to induce oncogenic translocations between *nuclear*

receptor coactivator 2 (*Ncoa2*) and *gene regulated by estrogen in breast cancer protein* (*Greb1*), as well as between *tyrosine 3-monooxygenase/tryptophan 5-monooxygenase activation protein epsilon* (*Ywhae*) and *NUT family member 2* (*Nutm2*). Both fusions are drivers of uterine sarcomas and represent clinically relevant targets for verifying the versatility of our system (Fig 5A, 5C) [34,35].

Crucially, to rigorously test for complex repair errors, we also re-examined a specific 7.67-Mb inversion previously reported in our embryo-based studies [14] (S2A Fig). In *Recql5*-mutated embryos, this locus exhibits "chromoanasynthesis-like" events characterized by extensive structural alterations—such as localized duplications and triplications—arising from replication-based mechanisms like fork stalling and template switching (FoSTeS) or microhomology-mediated break-induced

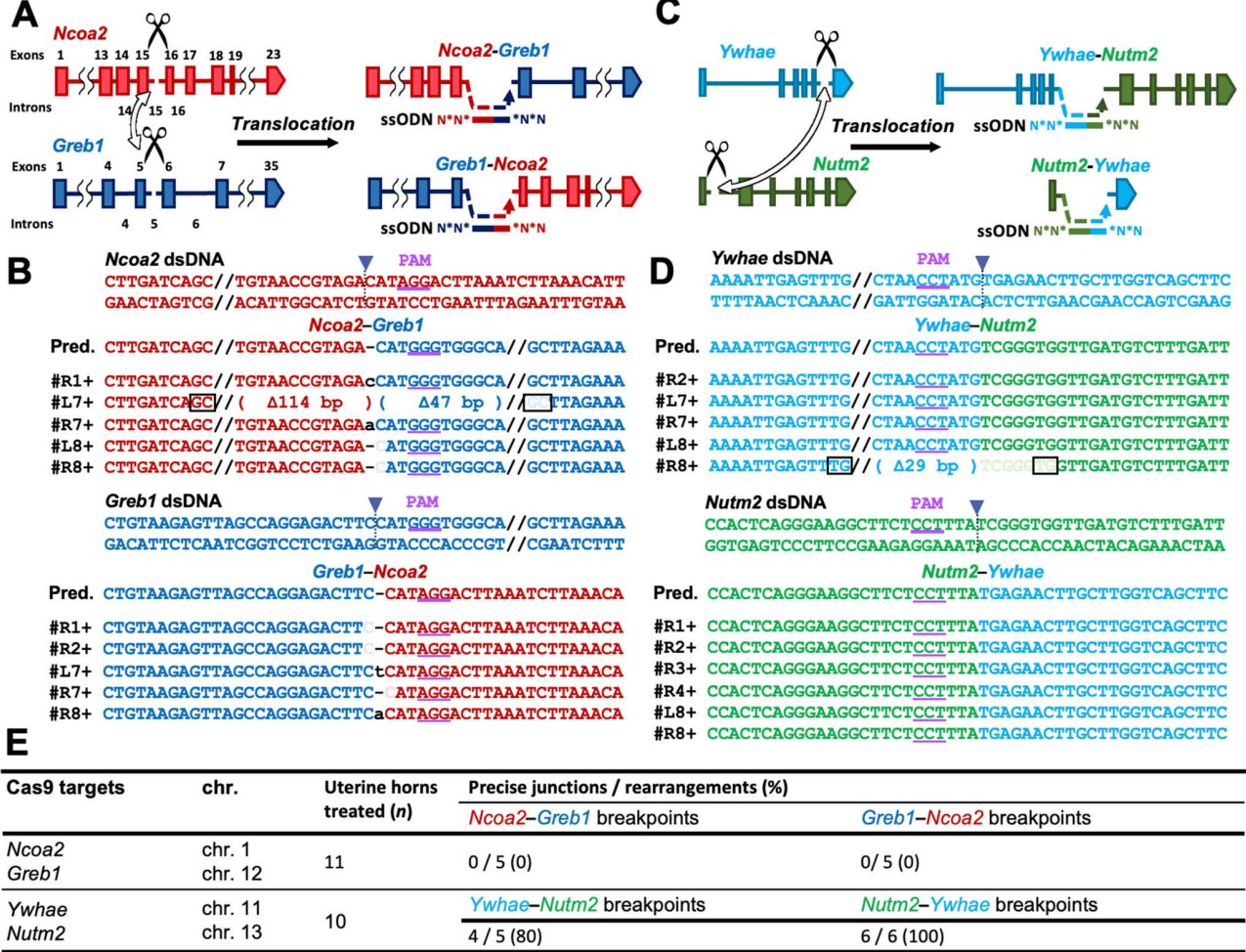

**Fig 5. Oncogenic translocations in the uterine epithelium induced via *in vivo* electroporation. (A)** Schematic of the translocation between *Ncoa2* (chr. 1) and *Greb1* (chr. 12) showing Cas9 cut sites (scissors) and ssODN donors for junction repair. **(B)** Sequence alignment of the PCR products corresponding to the genomic breakpoint junctions of *Ncoa2* and *Greb1*. The arrows indicate Cas9 cleavage sites; boxed bases, microhomology; lower-case letters, insertions or mismatches; underlines, PAM sequences. PAM, protospacer adjacent motif; dsDNA, double-stranded DNA; Pred., predicted sequence. **(C)** Schematic of the translocation between *Ywhae* (chr. 11) and *Nutm2* (chr. 13). **(D)** Sequence alignment of the PCR products corresponding to the genomic breakpoint junctions of *Ywhae* and *Nutm2*. **(E)** Translocation efficiency of *Ncoa2–Greb1* and *Ywhae–Nutm2* in the uterine epithelium. Precise junctions denote ssODN-consistent joins; rearrangements, all detected junctions connecting the intended loci.

replication (MMBIR). Targeting this "MMBIR-prone" locus allowed us to directly assess whether wild-type somatic cells repair DSBs with similar complexity or through simpler mechanisms.

Across all tested loci, the predicted rearrangement junctions were successfully detected by PCR (Fig 5B, 5D, 5E; S1D Fig; S2B, S2C Fig; S4 File). Most notably, sequence analysis revealed that the junctions were free of the complex signatures typically associated with FoSTeS/MMBIR, such as extended microhomology or localized copy number gains. This indicates that, unlike the chaotic repair observed in *Recql5*-mutated embryos, repair outcomes in somatic uterine epithelial cells via this method are predominantly defined by simple ligation events without complex structural errors.

Regarding the oncogenic fusions, although the genomic fusion events were successfully generated, no overt tumours were observed during the short observation period. Since tumour formation frequently requires extended latency, secondary mutations, or a higher clonal burden [36], we restricted our analysis here to the molecular validation of the rearrangement. Collectively, these findings demonstrate that *in vivo* electroporation is a robust platform for inducing and analyzing specific chromosomal rearrangements at multiple genetic loci, confirming its utility for generating somatic cancer-associated genotypes with high repair fidelity.

## Molecular identification of chromosomal rearrangements via whole-genome sequencing

To obtain molecular data on the chromosomal rearrangements induced by our approach, we performed whole-genome sequencing (WGS) on edited uterine tissues (Illumina NovaSeq6000, paired-end 150 bp, ~90 Gb per sample). A substantial challenge in analysing bulk tissue from this procedure is the dilution of the signal: genomic DNA was extracted from the entire uterus, whereas CRISPR RNP delivery was spatially restricted to the epithelial layer (Fig 1B). Consequently, the vast excess of unedited stromal and myometrial DNA diluted the variant allele frequency below the default thresholds of automated structural-variant callers like Manta [37].

To address this, we performed a targeted manual inspection of BAM files using samtools and Integrative Genomics Viewer (IGV) within ±2 Mb windows around the predicted loci. Through this focused analysis, we identified discordant read pairs that bridged the expected chromosomes. For instance, we detected four pairs linking chr15 and chr2 (*Ypel4N–Atf4N*), six pairs linking chr1 and chr12 (*Ncoa2–Greb1*), and three pairs linking chr11 and chr13 (*Ywhae–Nutm2*) (S4 and S5 Tables).

Analysis of these reads confirmed the structural linkage between the targeted chromosomes. The majority of these chimeric read pairs exhibited mapping quality (MAPQ) scores of 60—the maximum assigned by the aligner—indicating their unique placement in the genome. Importantly, these pairs spanned the expected fusion points: for each identified pair, one read mapped to the sequence flanking the cut site on the first chromosome, while its mate mapped to the corresponding region on the second chromosome. This specific bridging configuration provides molecular evidence that the two distinct loci have been physically joined.

We further estimated the relative abundance of rearranged alleles in the bulk DNA library by comparing discordant paired-end reads to properly paired reads. The apparent abundance was consistently low, ranging approximately from $3.4 \times 10^{-6}$ to $6.7 \times 10^{-6}$ (0.00034–0.00067%; S6 Table). This frequency is consistent with the expectations of somatic mosaicism combined with whole-organ tissue dilution.

Representative IGV screenshots (Fig 6) visualize these paired-end connections across the breakpoints. Taken together with the nested PCR and Sanger sequencing results, these WGS data provide molecular support for the presence of CRISPR/Cas9-induced chromosomal translocations within the uterine tissue. Although rare in the bulk context, the identification of these specific junctional reads supports the validity of our *in vivo* electroporation-based approach.

## Discussion

Using CRISPR RNP electroporation, we investigated the feasibility of inducing interchromosomal translocations and repairing a 57.8-Mb inversion in the mouse uterine epithelium. This study serves to establish a proof-of-concept that

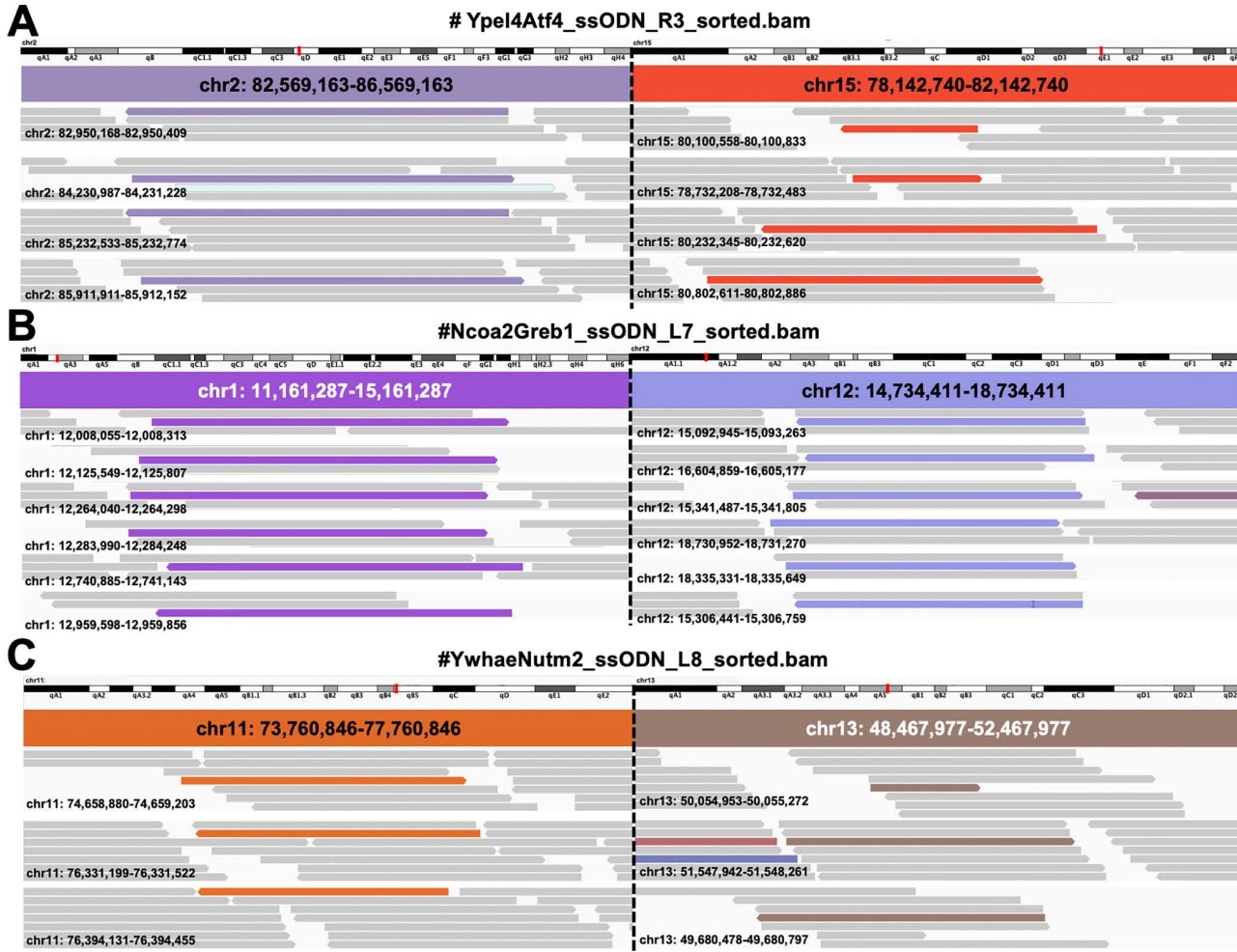

**Fig 6. IGV visualization of chromosomal translocations detected by whole-genome sequencing of edited uterine tissue. (A–C)** Integrative Genomics Viewer (IGV) screenshots showing paired-end read alignments spanning the predicted translocation breakpoints. **(A)** Translocation between chromosomes 2 (purple) and 15 (orange) at the *Ypel4N–Atf4N* locus. **(B)** Translocation between chromosomes 1 (purple) and 12 (blue) at the *Ncoa2–Greb1* locus. **(C)** Translocation between chromosomes 11 (orange) and 13 (brown) at the *Ywhae–Nutm2* locus. Discordant paired reads bridging the two chromosomes are color-coded by mate-pair orientation, and normal reads are shown in gray.

transient, virus-free genome editing can reconstruct or resolve large-scale chromosomal abnormalities *in vivo*. While the editing efficiency remains low due to the inherent constraints of somatic mosaicism, our findings suggest that non-viral approaches can complement existing technologies by offering a chemically defined platform for studying the mechanisms of chromosomal rearrangement.Compared with lentiviral or AAV-based systems, which represent the current gold standard for *in vivo* delivery [11,38], electroporation avoids vector production and prolonged Cas9 expression. We acknowledge that our current method yields significantly lower editing frequencies than viral vectors and requires microsurgical access. However, unlike viral methods that introduce integration/cargo complexities, our RNP approach offers "compositional simplicity" and a transient editing window. Therefore, rather than proposing this as a high-efficiency clinical alternative, we present it as a simplified experimental strategy for specific contexts—such as mechanistic analysis of repair outcomes—where a "clean" genomic background without viral integration is prioritized. Beyond the avoidance of viral integration, the use of transient RNP delivery offers several additional advantages over lentiviral or plasmid-based systems.

Specifically, the limited duration of Cas9 activity substantially reduces off-target editing accumulation, and mitigates the risk of catastrophic genomic events associated with prolonged nuclease expression, such as chromothripsis, large-scale deletions, aneuploidy, chromosome loss, p53-mediated enrichment of oncogenic cells, and integration of exogenous sequences including viral sequences, plasmids, and retrotransposons [39].

To detect and characterize these rare events, we strategically employed high-depth short-read whole-genome sequencing. Although long-read sequencing is advantageous for structural phasing, we selected short-read sequencing to ensure the ultra-deep coverage required for identifying rare somatic alleles ($10^{-6}$) and to provide the nucleotide-level resolution necessary for validating junctional fidelity. High-quality discordant read pairs bridged the expected loci (e.g., *Ypel4–Atf4*; Fig 6), confirming that the intended rearrangements occurred. We also acknowledge that targeted deep sequencing of regions surrounding CRISPR cutting sites, including amplicon-based approaches, would offer enhanced sensitivity for detecting rare or complex repair outcomes, such as the integration of retrotransposons at editing sites [40]. Such approaches could complement WGS in future studies aimed at more comprehensive characterization of editing outcomes.

The low allele abundance (approximately $3.4 \times 10^{-6}$–$6.7 \times 10^{-6}$) reflects the substantial dilution of edited epithelial cells by unedited stromal tissues. Given this rarity and the non-selected nature of the events, these rearrangements likely represent mono-allelic (heterozygous) modifications, effectively modeling the stochastic "first hit" scenarios often seen in early somatic tumorigenesis. These values indicate that the current efficiency limits the method's utility to applications where sensitive molecular detection is applicable, rather than those requiring high-level tissue modification.

Our sequence analysis highlighted the potential of this system to assess repair fidelity in a somatic context. We observed junctions restored via precise repair as well as those showing signatures of NHEJ/MMEJ. Unlike our previous observations in Recql5-mutated embryos, complex structural errors were absent, suggesting a distinct repair profile in these somatic cells. Thus, despite the low frequency of events, the ability to retrieve nucleotide-level information from *in vivo* samples demonstrates the value of this approach for dissecting repair pathways without the confounding factors of viral delivery. We acknowledge that Sanger sequencing of cloned PCR products, while sufficient for nucleotide-resolution verification of individual junction sequences, has inherent limitations in characterizing the full spectrum and frequency of editing outcomes. NGS-based amplicon sequencing would provide more comprehensive quantitative information regarding indel distributions and precise junction frequencies. Future studies employing such approaches will be necessary to fully characterize the repair outcome landscape at each targeted locus.

We also recognize the limitations regarding cytogenetic visualization. Although FISH and karyotyping are standard for visualizing rearrangements, the low proliferative index of the uterine epithelium and the rarity of edited cells made obtaining informative metaphase spreads impractical in this study. Consequently, we prioritized WGS and PCR, which provide the nucleotide-level resolution necessary to evaluate repair fidelity—information not accessible via standard cytogenetics. Future improvements in editing efficiency or enrichment strategies, such as the use of fluorescently tagged Cas9 for FACS-based isolation, will be necessary to enable direct cell-by-cell assessment.

In conclusion, this study demonstrates the fundamental feasibility of using CRISPR RNP electroporation for somatic chromosome engineering. The modification of a 57.8-Mb inversion underscores the potential to model large-scale genomic defects. Although substantial improvements in efficiency are required for broader application, our work provides a baseline for future studies aiming to link structural changes to repair mechanisms in a physiologically relevant, non-viral setting.

## Supporting information

**S1 Fig. Representative PCR genotyping of genomic DNA extracted from uterine horns edited by *in vivo* electroporation.**
(PDF)

**S2 Fig. Inversion in the uterine epithelium induced via *in vivo* electroporation.**
(PDF)

**S1 Table. gRNAs used in the present study.**
(PDF)

**S2 Table. Single-stranded oligodeoxynucleotides (ssODNs) used in the present study.**
(PDF)

**S3 Table. Primers used in the present study.**
(PDF)

**S4 Table. Discordant read pairs identified by whole-genome sequencing of edited uterine tissue.**
(PDF)

**S5 Table. Nucleotide sequences of discordant read pairs identified by whole-genome sequencing.**
(PDF)

**S6 Table. Summary of WGS-detected translocation-supporting read pairs and apparent junction frequencies.**
(PDF)

**S1 File. S1 Raw images. Uncropped and unadjusted gel images (related to S1A Fig).**
(PDF)

**S2 File. S2 Raw images. Uncropped and unadjusted gel images (related to S1B Fig).**
(PDF)

**S3 File. S3 Raw images. Uncropped and unadjusted gel images (related to S1C Fig).**
(PDF)

**S4 File. S4 Raw images. Uncropped and unadjusted gel images (related to S1D Fig).**
(PDF)

## Acknowledgments

The authors thank the laboratory members for their support with animal care and experiments. We would like to thank Editage (www.editage.jp) for English language editing.

## Author contributions

**Conceptualization:** Satoru Iwata.

**Funding acquisition:** Satoru Iwata.

**Investigation:** Satoru Iwata, Yumi Miura.

**Methodology:** Satoru Iwata.

**Project administration:** Satoru Iwata.

**Supervision:** Takashi Iwamoto.

**Writing – original draft:** Satoru Iwata.

**Writing – review & editing:** Takashi Iwamoto.

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
