## [Decision Letter · Decision Letter 0]

25 Mar 2026

PONE-D-26-02364

Non-viral in vivo electroporation-based chromosomal engineering and repair assessment in the murine uterine epithelium

PLOS One

Dear Dr. Iwata,

Thank you for submitting your manuscript to PLOS ONE. After careful consideration, we feel that it has merit but does not fully meet PLOS ONE’s publication criteria as it currently stands. Therefore, we invite you to submit a revised version of the manuscript that addresses the points raised during the review process.

Since none of the previous reviewers of Review Commons were available, I invited three new experts for reviewing your revised manuscript. As you find in their comments, their responses are positive, and they pointed out some minor issues. Please revised it according to their suggestions.

We look forward to receiving your revised manuscript.

Kind regards,

Hodaka Fujii, M.D., Ph.D.

Academic Editor

PLOS One

Journal Requirements:

2. 1) To comply with PLOS One submissions requirements, in your Methods section, please provide additional information regarding the experiments involving animals and ensure you have included details on (1) methods of sacrifice, (2) methods of analgesia, and (3) efforts to alleviate suffering."

2) You mention in your Methods that mice are euthanised one hour after electroporation, but that DNA sequencing takes place seven days postoperatively. In your Methods, please clarify how many mice were euthanised after one hour, and the total number of mice used.

5. Please provide a complete Data Availability Statement in the submission form, ensuring you include all necessary access information or a reason for why you are unable to make your data freely accessible. If your research concerns only data provided within your submission, please write "All data are in the manuscript and/or supporting information files" as your Data Availability Statement.

6. PLOS requires an ORCID iD for the corresponding author in Editorial Manager on papers submitted after December 6th, 2016. Please ensure that you have an ORCID iD and that it is validated in Editorial Manager. To do this, go to ‘Update my Information’ (in the upper left-hand corner of the main menu), and click on the Fetch/Validate link next to the ORCID field. This will take you to the ORCID site and allow you to create a new iD or authenticate a pre-existing iD in Editorial Manager.

7. We notice that your supplementary figures and tables are included in the manuscript file. Please remove them and upload them with the file type 'Supporting Information'. Please ensure that each Supporting Information file has a legend listed in the manuscript after the references list.

8. PLOS ONE now requires that authors provide the original uncropped and unadjusted images underlying all blot or gel results reported in a submission’s figures or Supporting Information files. This policy and the journal’s other requirements for blot/gel reporting and figure preparation are described in detail at https://journals.plos.org/plosone/s/figures#loc-blot-and-gel-reporting-requirements and https://journals.plos.org/plosone/s/figures#loc-preparing-figures-from-image-files. When you submit your revised manuscript, please ensure that your figures adhere fully to these guidelines and provide the original underlying images for all blot or gel data reported in your submission. See the following link for instructions on providing the original image data: https://journals.plos.org/plosone/s/figures#loc-original-images-for-blots-and-gels.

Reviewers' comments:

Reviewer's Responses to Questions

**Comments to the Author**

1. Is the manuscript technically sound, and do the data support the conclusions?

Reviewer #1: Yes

Reviewer #2: Yes

Reviewer #3: Yes

2. Has the statistical analysis been performed appropriately and rigorously?

Reviewer #1: Yes

Reviewer #2: Yes

Reviewer #3: I Don't Know

3. Have the authors made all data underlying the findings in their manuscript fully available?

Reviewer #1: Yes

Reviewer #2: Yes

Reviewer #3: Yes

4. Is the manuscript presented in an intelligible fashion and written in standard English?

Reviewer #1: Yes

Reviewer #2: Yes

Reviewer #3: Yes

5. Review Comments to the Author

Reviewer #1: This study describes a non-viral, Cas9 RNP–based electroporation approach to induce defined somatic chromosomal rearrangements in the murine uterine epithelium. The authors validate interchromosomal translocations and a large-scale inversion at nucleotide resolution using PCR and high-depth WGS. Despite low apparent allele frequencies due to mosaicism, the method provides a useful in vivo platform to study chromosomal engineering and DNA repair without viral delivery. This is a hot area. Though the efficiency is relatively low and methodology (sanger sequencing) is not very sufficient, overall the quality of the paper is good, with solid insights. I have few minor comments below.

1. The authors could polish these two sentences to make them connect more naturally "Understanding the mechanisms underlying the formation and repair of these abnormalities is crucial. The clustered regularly interspaced palindromic repeat (CRISPR)/CRISPR-associated protein (Cas) system has become a widely used tool for modelling these events [7], [8]."

2. Fig. 2B, it is highly recommend the authors to include with or without ssODN in the figure, instead of + or -. to facilitate the readership. And the authors should also demonstrate the limitation of using Sanger sequncing to evaluate the freqeuncy and characteristic of editing outputs, compared to NGS based comprehensive analysis. Moreover, I was not clear that why the authors use three CRISPR cutting sites to induce two inversions at the same time, as this will make the repair much more complex including large deletions (or one fragment deletion), complex inversions... The best senario should be only two DSBs make one translocation

3. Likewise, in Fig. 3e and 4c, it is difficult to conclude when there are only two target sequences.

4. besides Fig. 6, it would be necessary to show the sequences of these reads

5. Compared to WGS, personally I believe the targeted sequencing expanding the CRISPR cutting sites should work better to characterize the editing outcomes including translocation as well as unwanted edits (see PMID: 35760782)

6. Discussion, the advantage of RNP compared to duration lentivirus delivery includes that it could largely eliminate the CRISPR off-targets, as well as the unwanted outcomes including larger genomic rearrangements or even more catastrophic events such as chromothripsis, aneuploidy, chromosome loss and p53 activation that can enrich oncogenic cells, integrations of exogenous sequences (virus, plasmids, and retrotransposons) as reviewed in PMID: 36639728

Reviewer #2: In this study, Iwata et al. evaluated an in vivo electroporation-based approach to delivery Cas9 ribonucleoproteins (RNPs) in the murine uterine epithelium to generate diverse chromosomal rearrangements. Throughout the study, the authors used one standard approach to deliver Cas9 RNPs in the uterine lumen using electroporation to genetically manipulate the epithelium except different targeting guide RNAs/ single-stranded DNA oligonucleotides were varied for each intended chromosomal rearrangement/ genetic manipulation. The authors validate the genetic manipulations using PCR analysis of the edited tissue confirming that they indeed can generate diverse chromosomal rearrangements including large deletions, translocations, and inversions using their approach. Using WGS, the authors show that these chromosomal rearrangements as expected are rare in the uterine epithelium, demonstrating that this approach for in vivo genetic manipulation likely only results in a few cells being edited within the uterine epithelium. This is a key limitation of the approach of this study, which has been brought up as a criticism by two reviewers of the manuscript before. Nonetheless, the authors clearly acknowledge the limitation of their approach in this current version of the manuscript highlighting the rare frequency of the intended genetic manipulation within the uterine epithelium. Therefore, I suggest that the current version of the manuscript should be published since the authors clearly show that chromosomal rearrangements could be generated in the uterine epithelium with their reported approach, albeit at a rare frequency.

Reviewer #3: Although there are many CRISPR based methods and advances in germ line genome editing including large scale chromosomal modifications, the induction of defined chromosomal rearrangements directly in somatic tissues in vivo remains technically challenging and there are no reports demonstrating it thus far. The authors elegantly demonstrate this using in vivo electroporation which also overcomes the use of viral vectors.

The experiments are nicely done, and the data agrees with the interpretations and the overall conclusions. The manuscript is written well with as much technical details as possible, including extensive amount of background and rationale for the experimental strategies. One minor concern is the too low efficiency, which in my opinion, is not at all surprising considering the technical challenges in achieving chromosomal rearrangements in vivo. Nevertheless, this proof of principle study lays a foundation for future technological breakthroughs.

The data, figures, presentation and interpretations are highly commendable. One suggestion is that authors may consider testing the figure 1B experiment without applying electroporation. That is, imaging dextran fluorescence in oviductal epithelium after administering all components except applying electroporation.

6. PLOS authors have the option to publish the peer review history of their article (what does this mean?). If published, this will include your full peer review and any attached files.

**Do you want your identity to be public for this peer review?** For information about this choice, including consent withdrawal, please see our Privacy Policy.

Reviewer #1: No

Reviewer #2: No

Reviewer #3: **Yes:** CB Gurumurthy

---

## [Author Response · Author response to Decision Letter 1]

13 Apr 2026

・Response to Journal Requirements

We have addressed all journal requirements as follows:

1. The manuscript has been reformatted to comply with PLOS ONE style requirements.

2. We have added the following information to the Methods section to comply with PLOS ONE submission requirements:

(1) Methods of sacrifice: Mice used for the dextran tracer experiment (n=3) were euthanised by cervical dislocation at 1 hour post-electroporation. Mice used for genome editing experiments (n=50 in total) were euthanised by cervical dislocation at the experimental endpoint (≥7 days post-electroporation).

(2) Efforts to alleviate suffering: Following surgery, mice were placed on a warming pad maintained at 38°C overnight to support postoperative recovery. The reversible anaesthesia protocol employing medetomidine/midazolam/butorphanol, with postoperative reversal by atipamezole hydrochloride, was used to minimise the duration and depth of anaesthesia. All efforts were made to minimise animal suffering in accordance with the 3Rs principles.

3. Funding information has been removed from the manuscript text.

4. Grant numbers in the Funding Information and Financial Disclosure sections have been reconciled.

5. The Data Availability Statement has been completed, including the DDBJ SRA accession number (PRJDB37706).

6. The ORCID iD of the corresponding author has been validated in Editorial Manager.

7. Supplementary figures and tables have been separated from the manuscript and uploaded as Supporting Information files, with legends listed after the reference list.

8. Original uncropped and unadjusted gel images are provided as S1–S4 Raw images in Supporting Information.

9. Recommended references [39] [40] have been evaluated and cited where appropriate in the Discussion.

10. The reference list has been reviewed. Reference 10 (Maddalo et al., Nature 2014) has an associated Corrigendum (Nature 524, 502, 2015; DOI: 10.1038/nature14571), in which a schematic diagram in Fig. 1b was corrected. This correction does not affect the findings cited in our manuscript, and the reference has been retained. No retracted articles were identified in the reference list.

・Description of the revisions that have already been incorporated in the transferred manuscript

Reviewer #1

This study describes a non-viral, Cas9 RNP–based electroporation approach to induce defined somatic chromosomal rearrangements in the murine uterine epithelium. The authors validate interchromosomal translocations and a large-scale inversion at nucleotide resolution using PCR and high-depth WGS. Despite low apparent allele frequencies due to mosaicism, the method provides a useful in vivo platform to study chromosomal engineering and DNA repair without viral delivery. This is a hot area. Though the efficiency is relatively low and methodology (sanger sequencing) is not very sufficient, overall the quality of the paper is good, with solid insights. I have few minor comments below.

General Response:

We sincerely thank Reviewer #1 for the thorough and encouraging assessment of our manuscript. We are pleased that the reviewer recognized the value of this work as a useful in vivo platform for studying chromosomal engineering and DNA repair in a non-viral setting. We fully agree that the relatively low editing efficiency and the limitations of Sanger sequencing are important points, and we have addressed these concerns in detail in our point-by-point responses below.

Comment 1:

The authors could polish these two sentences to make them connect more naturally "Understanding the mechanisms underlying the formation and repair of these abnormalities is crucial. The clustered regularly interspaced palindromic repeat (CRISPR)/CRISPR-associated protein (Cas) system has become a widely used tool for modelling these events [7], [8]."

Revised Response:

We thank the reviewer for this suggestion. We have revised the two sentences to read more naturally as follows (lines 46–49):

"Understanding the mechanisms underlying the formation and repair of these abnormalities is crucial, and the clustered regularly interspaced palindromic repeat (CRISPR)/CRISPR-associated protein (Cas) system has become a widely used tool for modelling and investigating such events [7], [8]."

Comment 2:

Fig. 2B, it is highly recommend the authors to include with or without ssODN in the figure, instead of + or -. to facilitate the readership. And the authors should also demonstrate the limitation of using Sanger sequncing to evaluate the freqeuncy and characteristic of editing outputs, compared to NGS based comprehensive analysis. Moreover, I was not clear that why the authors use three CRISPR cutting sites to induce two inversions at the same time, as this will make the repair much more complex including large deletions (or one fragment deletion), complex inversions... The best senario should be only two DSBs make one translocation

Revised Response:

We thank the reviewer for these detailed and constructive comments. We address each point separately below.

(1) Figure label revision: "+" and "−" → "with ssODN" and "without ssODN"

We thank the reviewer for this helpful suggestion. To improve readability, we have added explicit group headers "with ssODN" and "without ssODN" directly within the sequence alignment panels of Fig 2B, Fig 4B, and all related figures, clearly separating the two experimental conditions while preserving the individual sample identifiers.

(2) Limitation of Sanger sequencing compared to NGS-based analysis

We acknowledge that Sanger sequencing of cloned PCR products has inherent limitations in evaluating the frequency and full spectrum of editing outcomes. Specifically, it captures only a limited number of alleles and cannot provide the comprehensive quantitative information obtainable by NGS-based amplicon sequencing. We have added the following statement to the Discussion section (lines 594–600):

"We acknowledge that Sanger sequencing of cloned PCR products, while sufficient for nucleotide-resolution verification of individual junction sequences, has inherent limitations in characterizing the full spectrum and frequency of editing outcomes. NGS-based amplicon sequencing would provide more comprehensive quantitative information regarding indel distributions and precise junction frequencies. Future studies employing such approaches will be necessary to fully characterize the repair outcome landscape at each targeted locus."

(3) Rationale for three simultaneous CRISPR cutting sites

We appreciate the reviewer's concern regarding the complexity introduced by using three simultaneous cut sites. We wish to clarify that this design was not intended to induce two independent inversions simultaneously, but rather to generate a complex chromosomal rearrangement (CCR) involving three breakpoints on chromosome 10, recapitulating the type of multi-breakpoint rearrangements observed in human cancers and previously modelled in our embryo-based study [14]. The use of three simultaneous cut sites was intentionally designed to assess whether our electroporation platform can handle the complexity of multi-breakpoint rearrangements in a somatic context. We have added the following clarification to the Results section (lines 294–299):

"This locus is implicated in various human cancers and shares a syntenic configuration with human chromosome 12 [27] (Fig 2A), and was previously targeted in our embryo-based studies [14]. The simultaneous use of three cut sites was intentionally designed to recapitulate multi-breakpoint CCRs and to assess whether our electroporation platform can handle such complexity in a somatic context."

We also note that the subsequent translocation experiments each employed only two cut sites per rearrangement event, demonstrating that our platform is equally applicable to simpler two-DSB designs.

Comment 3:

Likewise, in Fig. 3e and 4c, it is difficult to conclude when there are only two target sequences.

Revised Response:

We thank the reviewer for this observation. We interpret this comment as reflecting concern that the small number of sequence-verified precise junctions at individual loci in Fig 3E and 4C limits the confidence of conclusions drawn regarding precise repair frequency. We have made the following specific revisions to address this concern.

(1) Results section — Fig 3 paragraph (lines 362–369):

"These results demonstrate that interchromosomal rearrangements can be generated in vivo using this electroporation-based approach. Given the limited number of recoverable rearrangement-positive clones per locus, the proportions of precise junctions shown in Fig 3E are presented as qualitative indicators of repair outcome categories, consistent with the proof-of-concept scope of this study."

(2) Results section — Fig 4 paragraph (lines 409–416):

"Under ssODN(+) conditions, precise restoration was confirmed in 1 of 2 sequenced amplicons at the left breakpoint and 3 of 4 at the right breakpoint. In contrast, while bridging of the breakpoints was detected in ssODN(−) experiments, they lacked the precise sequence restoration observed with donor templates (Fig 4B, C; S1C Fig). Although the number of sequence-verified clones is limited, the complete absence of precise junctions in ssODN(−) experiments suggests that the provision of ssODNs can steer the repair machinery towards precise ligation at these distant breakpoints."

These revisions ensure that the data in Fig 3E and 4C are presented as qualitative evidence of feasibility rather than statistically robust estimates of precise repair frequency, consistent with the proof-of-concept framing of this study.

Comment 4:

besides Fig. 6, it would be necessary to show the sequences of these reads

Revised Response:

We thank the reviewer for this suggestion. To address this concern, we have added the actual nucleotide sequences of each discordant read pair to S5 Table. This allows readers to directly inspect the sequence-level evidence for each translocation junction identified by WGS, complementing the IGV visualization shown in Fig 6.

Comment 5:

Compared to WGS, personally I believe the targeted sequencing expanding the CRISPR cutting sites should work better to characterize the editing outcomes including translocation as well as unwanted edits (see PMID: 35760782)

Revised Response:

We thank the reviewer for pointing out the significance of targeted sequencing, especially in the context of recent findings such as LINE-1 insertions [PMID: 35760782]. We agree that targeted deep sequencing of regions surrounding CRISPR cutting sites, such as the amplicon sequencing approach employed by Tao et al. [40], offers higher sensitivity for detecting rare unwanted editing outcomes including retrotransposon insertions. This approach would complement WGS in future studies aimed at more comprehensive safety assessment of the editing outcomes.

While our current study focused on a broader, unbiased overview using WGS to establish the proof-of-concept for this in vivo method, we fully acknowledge the value of the suggested approach for future high-resolution safety assessments. As suggested, we have added a sentence to the Discussion citing the recommended study and highlighting targeted NGS as a crucial tool for future investigation.

Revision to Discussion (lines 574–579):

"We also acknowledge that targeted deep sequencing of regions surrounding CRISPR cutting sites, including amplicon-based approaches, would offer enhanced sensitivity for detecting rare or complex repair outcomes, such as the integration of retrotransposons at editing sites [40]. Such approaches could complement WGS in future studies aimed at more comprehensive characterization of editing outcomes."

Comment 6:

Discussion, the advantage of RNP compared to duration lentivirus delivery includes that it could largely eliminate the CRISPR off-targets, as well as the unwanted outcomes including larger genomic rearrangements or even more catastrophic events such as chromothripsis, aneuploidy, chromosome loss and p53 activation that can enrich oncogenic cells, integrations of exogenous sequences (virus, plasmids, and retrotransposons) as reviewed in PMID: 36639728

Revised Response:

We thank the reviewer for this insightful suggestion and for providing the relevant reference [PMID: 36639728]. We have expanded the Discussion to include the following (lines 560–567):

"Beyond the avoidance of viral integration, the use of transient RNP delivery offers several additional advantages over lentiviral or plasmid-based systems. Specifically, the limited duration of Cas9 activity substantially reduces off-target editing accumulation, and mitigates the risk of catastrophic genomic events associated with prolonged nuclease expression, such as chromothripsis, large-scale deletions, aneuploidy, chromosome loss, p53-mediated enrichment of oncogenic cells, and integration of exogenous sequences including viral sequences, plasmids, and retrotransposons [39]."

Reviewer #2

In this study, Iwata et al. evaluated an in vivo electroporation-based approach to delivery Cas9 ribonucleoproteins (RNPs) in the murine uterine epithelium to generate diverse chromosomal rearrangements. Throughout the study, the authors used one standard approach to deliver Cas9 RNPs in the uterine lumen using electroporation to genetically manipulate the epithelium except different targeting guide RNAs/ single-stranded DNA oligonucleotides were varied for each intended chromosomal rearrangement/ genetic manipulation. The authors validate the genetic manipulations using PCR analysis of the edited tissue confirming that they indeed can generate diverse chromosomal rearrangements including large deletions, translocations, and inversions using their approach. Using WGS, the authors show that these chromosomal rearrangements as expected are rare in the uterine epithelium, demonstrating that this approach for in vivo genetic manipulation likely only results in a few cells being edited within the uterine epithelium. This is a key limitation of the approach of this study, which has been brought up as a criticism by two reviewers of the manuscript before. Nonetheless, the authors clearly acknowledge the limitation of their approach in this current version of the manuscript highlighting the rare frequency of the intended genetic manipulation within the uterine epithelium. Therefore, I suggest that the current version of the manuscript should be published since the authors clearly show that chromosomal rearrangements could be generated in the uterine epithelium with their reported approach, albeit at a rare frequency.

General Response:

We sincerely thank Reviewer #2 for their thorough and positive assessment of the revised manuscript. We are pleased that the addition of WGS analysis, the transparent reporting of editing frequency, and the proof-of-concept framing adequately addressed the concerns raised in the previous review. We have made no substantive changes in response to Reviewer #2's comments, as they recommended publication of the current version.

Reviewer #3

Although there are many CRISPR based methods and advances in germ line genome editing including large scale chromosomal modifications, the induction of defined chromosomal rearrangements directly in somatic tissues in vivo remains technically challenging and there are no reports demonstrating it thus far. The authors elegantly demonstrate this using in vivo electroporation which also overcomes the use of viral vectors.

The experiments are nicely done, and the data agrees with the interpretations and the overall conclusions. The manuscript is written well with as much technical details as possible, including extensive amount of background and rationale for the experimental strategies. One minor concern is the too low efficiency, which in my opinion, is not at all surprising considering the technical challenges in achieving chromosomal rearrangements in vivo. Nevertheless, this proof of principle study lays a foundation for future technological breakthroughs.

The data, figures, presentation and interpretations are highly commendable. One suggestion is that authors may consider testing the figure 1B experiment without applying elec

---

## [Decision Letter · Decision Letter 1]

21 Apr 2026

Non-viral in vivo electroporation-based chromosomal engineering and repair assessment in the murine uterine epithelium

PONE-D-26-02364R1

Dear Dr. Iwata,

We’re pleased to inform you that your manuscript has been judged scientifically suitable for publication and will be formally accepted for publication once it meets all outstanding technical requirements.

Kind regards,

Hodaka Fujii, M.D., Ph.D.

Academic Editor

PLOS One

Additional Editor Comments (optional):

Reviewers' comments:

Reviewer's Responses to Questions

**Comments to the Author**

1. If the authors have adequately addressed your comments raised in a previous round of review and you feel that this manuscript is now acceptable for publication, you may indicate that here to bypass the “Comments to the Author” section, enter your conflict of interest statement in the “Confidential to Editor” section, and submit your "Accept" recommendation.

Reviewer #1: All comments have been addressed

2. Is the manuscript technically sound, and do the data support the conclusions?

Reviewer #1: Yes

3. Has the statistical analysis been performed appropriately and rigorously?

Reviewer #1: Yes

4. Have the authors made all data underlying the findings in their manuscript fully available?

Reviewer #1: Yes

5. Is the manuscript presented in an intelligible fashion and written in standard English?

Reviewer #1: Yes

6. Review Comments to the Author

Reviewer #1: All my questions have been answered properly, and I do not have other concerns. Congrats the authors!

7. PLOS authors have the option to publish the peer review history of their article (what does this mean?). If published, this will include your full peer review and any attached files.

Reviewer #1: No

---

## [Editor Report · Acceptance letter]

PONE-D-26-02364R1

PLOS One

Dear Dr. Iwata,

I'm pleased to inform you that your manuscript has been deemed suitable for publication in PLOS One. Congratulations! Your manuscript is now being handed over to our production team.

Kind regards,

on behalf of

Dr. Hodaka Fujii

Academic Editor

PLOS One